# Corrosion Behavior and Biocompatibility of Hot-Extruded Mg–Zn–Ga–(Y) Biodegradable Alloys

**DOI:** 10.3390/jfb13040294

**Published:** 2022-12-12

**Authors:** Viacheslav Bazhenov, Anna Li, Artem Iliasov, Vasily Bautin, Sofia Plegunova, Andrey Koltygin, Alexander Komissarov, Maxim Abakumov, Nikolay Redko, Kwang Seon Shin

**Affiliations:** 1Casting Department, National University of Science and Technology “MISiS”, Leninskiy pr. 4, 119049 Moscow, Russia; 2Laboratory of Hybrid Nanostructured Materials, National University of Science and Technology “MISiS”, Leninskiy pr. 4, 119049 Moscow, Russia; 3Laboratory of Biomedical Nanomaterials, National University of Science and Technology “MISiS”, Leninskiy pr. 4, 119049 Moscow, Russia; 4Department of Metallurgy Steel, New Production Technologies and Protection of Metals, National University of Science and Technology “MISiS”, Leninskiy pr. 4, 119049 Moscow, Russia; 5Laboratory of Medical Bioresorption and Bioresistance, Moscow State University of Medicine and Dentistry, Delegatskaya 20/1, 127473 Moscow, Russia; 6Magnesium Technology Innovation Center, Department of Materials Science and Engineering, Seoul National University, 1 Gwanak-ro, Gwanak-gu, Seoul 08826, Korea

**Keywords:** biomaterials, corrosion resistance, cytotoxicity, gallium, magnesium, Hanks’ solution

## Abstract

Fixation screws and other temporary magnesium alloy fixation devices are used in orthopedic practice because of their biodegradability, biocompatibility and acceptable biodegradation rates. The substitution of dissolving implant by tissues during the healing process is one of the main requirements for biodegradable implants. Previously, clinical tests showed the effectiveness of Ga ions on bone tissue regeneration. This work is the first systematic study on the corrosion rate and biocompatibility of Mg–Zn–Ga–(Y) alloys prepared by hot extrusion, where Ga is an additional major alloying element, efficient as a bone-resorption inhibitor. Most investigated alloys have a low corrosion rate in Hanks’ solution close to ~0.2 mm/year. No cytotoxic effects of Mg–2Zn–2Ga (wt.%) alloy on MG63 cells were observed. Thus, considering the high corrosion resistance and good biocompatibility, the Mg–2Zn–2Ga alloy is possible for applications in osteosynthesis implants with improved bone tissue regeneration ability.

## 1. Introduction

A new paradigm in osteosynthesis is the use of biodegradable implants that would gradually dissolve as the healing process progresses [1,2]. This is because permanent titanium systems, which were the gold standard in osteosynthesis for many years, have a variety of disadvantages. These disadvantages include temperature sensitivity, tactile sensation, growth restrictions, imaging in radiotherapy, etc. [3]. The symptomatic removal of titanium implants in up to 40% of cases also have a negative impact on their use [3].

The appliance of temporary biodegradable implants helps minimizing the surrounding tissue inflammation induced by the implant, and avoids the need for secondary surgery for implant removal. Mg alloys are attractive candidates for temporary fixation devices for osteosynthesis due to good biocompatibility, satisfactorily high mechanical properties, and an acceptable biodegradation rate [4,5]. Furthermore, unlike Ti alloys, Mg alloys have a density and Young’s modulus close to those of cortical bone [6,7].

In comparison with permanent implants, biodegradable implants must guarantee the close rates of implant degradation and growth of bone tissue for substituting the voids in degraded implants with new bone tissue. It was established previously that Mg has a positive effect on the bone regeneration [8]. However, the addition of special components reinforcing bone growth to the alloy or to coating can increase the effectiveness of bone healing [9]. Research on animals revealed that presence of Ga in hydroxyapatite coating on Gription™ implants (West Chester, PA, USA) increased the rate of bone growth twice [10]. This is because Ga is well-known bone resorption inhibitor [11,12] and it effectively treats osteoporosis [13], hypercalcemia [14,15,16], Paget’s disease [17,18], and multiple myeloma [19]. Collery et al. reported the anti-cancer activity of gallium [20].

The positive effect of coating on the bone growth is temporary and lasts until the coating dissolves. In opposite, when one or several alloy components (Ga for example) improve the bone tissue growth process, the effectiveness of tissue healing is higher. Thus, the use of Ga as an alloying element for Mg biodegradable alloy can improve bone tissue regeneration ability. Provided benefits suggest that Mg alloy with Ga addition can be used for orthopedic implants with improved bone tissue regeneration ability.

One of the main problems in using both permanent and temporary implants is implant-related infection or inflammation. Zn and Ga have an antibacterial effect that can help inflammation prevention [21,22,23,24]. On the other hand, metals that have antibacterial effects demonstrate very strong cytotoxicity and poor biocompatibility [24]. The cytotoxicity analysis of Mg–Ga alloys showed that Ga at concentrations below 310 ng/mL did not show cytotoxicity [25].

The biodegradation rate is a critical parameter to provide the substitution of the voids in degraded implant by bone tissue. The investigation of corrosion properties of Mg–Ga alloys showed that increasing the Ga content in α-Mg resulted in a higher corrosion resistance, but when the Ga content is high and Ga presents in the form of the Mg_5_Ga_2_ intermetallic phase, the corrosion rate (CR) increases [25,26,27]. The heat treatment provides the dissolution of Mg_5_Ga_2_ and decreases the CR [28].

The Mg–4Zn–4Ga (wt.%) alloys, which alloyed by small quantities of Ca, Y, and Nd, processed by equal channel angular pressing (ECAP), were investigated in our previous work, and these alloys demonstrated high strength (up to 300 MPa) and a low CR of ~0.2 mm/year in Hanks’ solution [29]. In addition, it was established that Y does not decrease the CR of Mg–Zn–Ga alloys compared to Ca and Nd [29]. Therefore, the Mg–Zn–Ga alloy with Y addition needs to be investigated. The microstructure, phase composition, and mechanical properties of the five Mg–Zn–Ga–(Y) alloys after casting, heat-treatment, and hot extrusion, which are the subjects for this study, can be found in our previous work [30]. The best combination of mechanical properties is observed for the MgZn4Ga4 alloy after extrusion at 150 °C. The mentioned alloy show the following tensile properties: tensile yield strength (TYS) = 256 MPa, ultimate tensile strength (UTS) = 343 MPa, and elongation at fracture (El) = 14.2%. The compressive yield strength (CYS) and compressive strength (CS) for MgZn4Ga4 alloy were 251 and 587 MPa, respectively [30]. These values were on par with or better than those of several Mg alloys considered as candidate materials for bone fixation implants [29,31].

The objective of the study was to investigate the effect of the composition and extrusion temperature on the corrosion properties and biocompatibility of Mg–Zn–Ga–(Y) alloys that can provide the increased rate of bone tissue growth and provide substitution of the voids in the bone tissue by new generated bone. The present work is the first systematic study about the properties of Mg–Zn–(Y) alloys prepared by the hot extrusion process that contain Ga as a major alloying element in order to estimate its potential for utilization in orthopedic implants with improved bone-tissue regeneration ability.

## 2. Materials and Methods

### 2.1. Alloy Preparation and Hot Extrusion

Five Mg–Zn–Ga–(Y) alloys, with Zn and Ga contents listed in Table 1, were prepared. The chemical compositions of the alloys were analyzed using energy-dispersive X-ray spectroscopy (EDS) on the three areas with size 1 × 1 mm^2^. The high-purity charge materials, including Mg (99.98 wt.% purity; SOMZ, Solikamsk, Russia), Zn (99.995 wt.%; UGMK, Verkhnaya Pyshma, Russia), Ga (99.9999 wt.%; Girmet Ltd., Moscow, Russia), and Mg–20Y (wt.%) master alloy (Metagran, Moscow, Russia) were used for the preparation of alloys. For melt preparation, a steel crucible coated with a BN layer was used. Melt was prepared in a resistance furnace and was protected with an Ar + 2 vol.% SF_6_ gas mixture. Before pouring, the melt was purged with Ar for 3 min and held for 10 min. Ingots with a diameter of 60 mm and a length of 200 mm were obtained by casting in a preheated up to 150 °C aluminum permanent mold.

A two-stage solid solution heat treatment at 300 °C for 15 h + 400 °C for 30 h was applied for ingot homogenization. After heat treatment, ingots were machined into cylindrical billets of 145 mm height and 50 mm diameter. The 20 mm bars were obtained by direct extrusion (extrusion ratio of 6.25) at a ram speed of 1 mm/s on a 300-ton vertical hydraulic press PS-300A7 (Gidrosfera, Moscow, Russia) with a die preheated to 200–250 °C. Billet extrusion temperatures were 150, 200, and 250 °C [30].

### 2.2. Microstructural Observations and Phase Composition

The microstructure and phases elemental content were examined with the help of scanning electron microscopy (SEM; Vega SBH3, Tescan, Brno, Czech Republic) with an EDS detector (Oxford, UK).

### 2.3. Corrosion Testing

For in vitro immersion corrosion tests, samples 15 mm in diameter and 5 mm height, with a surface area of ~6 cm^2^ were used, made from pressed rods using electric spark wire-cutting. The finishing surface treatment was carried out by grinding on SiC sandpaper with a grain size of 320. The tests were carried out with exposure for 192 h at 37 °C in 400 mL of Hanks’ solution (PanEco, Moscow, Russia). At the same time, the ratio of the solution volume to the surface area of the sample was not less than 70 mL/cm^2^. In accordance with the ASTM standard [32], the calculation of the average CR (mm/year) was carried out, where the weight loss of the samples was determined from the volume of H_2_ released during the test [25,33,34]. Changes in the activity of hydrogen ions (pH) were measured using a HI83141 pH meter (Hanna Instruments, Woonsocket, PA, USA). For measurements, three to five samples were used for each alloy and extrusion temperature.

Measurements of the electrochemical corrosion of alloys were carried out in a three-electrode cell in Hanks’ solution at 37 °C using an IPC Pro MF potentiostat/galvanostat (Volta, St. Petersburg, Russia). In the measurement process, an alloy sample was used as a working electrode (exposure area 1 cm^2^), as well as silver/silver chloride (Ag/AgCl) reference electrodes and platinum as an auxiliary electrode. Alloy samples before electrochemical analysis were kept in an aqueous solution of HNO_3_ with a concentration of 0.3 wt.% for 2 s followed by washing in distilled water. Potentiodynamic polarization was performed at a scan rate of 1 mV/s from the cathode region −2.3 V to the anode region −1 V. Three to five samples for each alloy and extrusion temperature were used for measurements. Electrochemical process parameters, corrosion potential (*E_corr_*), corrosion current density (*i_corr_*), anodic and cathodic Tafel slopes, were determined by extrapolation of the Tafel region from polarization curves. The corrosion current densities used for the CR of alloys calculation [35].

### 2.4. Cytotoxicity Tests

A cell culture method following ISO 10993-5-2009 was used for cytotoxicity determination. Disk samples with a diameter of 15 mm and a height of 5 mm were obtained by wire cutting and ground using 320 grit abrasive SiC paper. In addition, a different line of samples taken after the immersion corrosion test (192 h immersion in Hanks’ solution) was used. Before the extraction procedure, the samples were sterilized in dry heat at 150 °C for 2 h. Then, the extracts were prepared by incubation of samples in Dulbecco’s Modified Eagle Medium (DMEM)/F12 cell medium (Sigma-Aldrich, Burlington, VE, USA) under standard cell culture conditions (37 °C and under 5% CO_2_ humidified sterile environment) for 12, 24, 72, or 168 h. The volume of the cell medium to the sample surface area ratio was 1 mL/cm^2^. A 4200-microwave plasma atomic emission spectrometer (Agilent, Santa Clara, CA, USA) was used for Mg, Zn, and Ga ion concentration measurements in extracts.

The MG63 human osteosarcoma cell line (American Type Culture Collection, Manassas, VI, USA) was cultured in a 5% CO_2_ atmosphere at 37 °C in DMEM/F12 medium containing 10% fetal bovine serum (Sigma-Aldrich, Burlington, VE, USA), 1% L-glutamine (Gibco, Waltham, MA, USA), and 1% antibiotics (penicillin and streptomycin).

The extract cytotoxicity was checked in vitro using the methyl tetrazolium salt (MTS) test, where the dye (3-(4,5-dimethylthiazol-2-yl)-5-(3-carboxymethoxyphenyl)-2-(4-sulfophenyl)-2H-tetrazolium) is added to the well and is processed by the cells into a formazan product. The dye and product have different absorption wavelengths at 490 nm. Therefore, it is possible to determine the proportion of viable cells as the ratio of the absorption coefficients of the sample to the control (*A_sample_*/*A_contr_*⋅100%), as measured using a spectrophotometer.

MG63 cells were seeded in 96-well plates at a concentration of 15,000/well. Their concentration was determined using an automatic cell counter (EVE, NanoEnTec Inc., Seoul, South Korea). After 24 h, the cell medium in the wells was partially or completely replaced with the extract. Titration was performed as follows, where 100%, 50%, 25%, and 12.5% of the cell medium was replaced with the extract. The cells were then incubated with the extract for 48 h and then MTS dye was added to them. Cells without the extract were used as negative controls. Cells treated with 20% ethanol were used as a positive control. Experiments were performed in triplicate.

## 3. Results

### 3.1. Corrosion Properties

Figure 1a–c shows the results of immersion corrosion tests of the Mg–Zn–Ga–(Y) alloys produced at different extrusion temperatures. The amount of hydrogen released over time for the alloy samples immersed in Hanks’ solution was not constant and decreased over time for all investigated alloys, except for MgZn6.5Ga2, which had a nearly constant hydrogen release rate. For example, for low-alloyed MgZn4Ga2 and MgZn2Ga2 alloys, the CR decreased significantly after the first 24 h of immersion, possibly due to the growth of a corrosion product layer that limits the contact between the alloy sample surfaces and the corrosive media, resulting in a protective effect.

The amount of hydrogen released during the 192 h immersion corrosion test was used to calculate the average CR following the G1-03 ASTM standard [32]. The CRs of the Mg–Zn–Ga–(Y) alloys at different hot extrusion temperatures are shown in Figure 2a. The highest CR of ~5 mm/year was observed for the MgZn6.5Ga2 alloy, which was attributed to the significant fraction of intermetallic phase remaining in the alloy structure after HT [30]. According to the XRD analysis [30], the predominant phase in MgZn6.5Ga2 after hot extrusion was MgZn. Therefore, it had a significant effect on the alloy corrosion behavior, acting as the cathode for α-Mg.

Meanwhile, all other alloys (extruded at 150 or 250 °C) showed a low CR of <0.5 mm/year. At the same extrusion temperatures for the MgZn4Ga4, MgZn4Ga2, and MgZn2Ga2 alloys, the lowest CR of ~0.2 mm/year was observed, which is close to that obtained for MgZn4Ga4 after ECAP (0.16 mm/year) [29]. The addition of Y to the MgZn4Ga4 alloy had little effect on the CR. It was difficult to evaluate the influence of extrusion temperature on the CR, because for MgZn4Ga4 and MgZn2Ga2 alloys, the highest CR was observed after extrusion at 200 °C. In contrast, for the MgZn4Ga2 alloy, the CR was higher after extrusion at temperature of 150 °C. Therefore, no significant relationship was observed between CR and extrusion temperature considering the confidence limit.

The pH of the corrosive media as a function of the extrusion temperature of Mg–Zn–Ga–(Y) alloys measured during the immersion corrosion tests is shown in Figure 2b. Considering the high uncertainty of the pH measurements, the pH of the Hanks’ solution was 7.5–8.5 for most alloy samples, which is close to that of the Hanks’ solution pH before the corrosion tests (7.4). However, when corrosion of MgZn6.5Ga2 alloys occurred, the pH at the end of the immersion corrosion test was 8.9–9.3, attributed to the release of hydroxide ions [33]. The observation of both a high CR of the alloys and a high pH of the corrosive medium is consistent with previous results [36,37].

Typical polarization curves obtained in Hanks’ solution at 37 °C for the Mg–Zn–Ga–(Y) alloys extruded at different temperatures are shown in Figure 3a–c. The polarization curves were similar for all extrusion temperatures. The variations in the anodic Tafel slopes for samples extruded at 200 °C was low and attributed to the different rates of inhibition of the anodic process of electrochemical corrosion, which is associated with the formation of intermetallic compounds [38,39]. However, the cathodic current differed significantly, and the maximum and minimum cathodic currents were observed for MgZn6.5Ga2 and MgZn2Ga2, respectively. Previously it was shown that with increasing of Zn and Ga content in alloy the summarized fraction of MgZn and Mg_5_Ga_2_ phases increased (form 0 vol.% for MgZn2Ga2 to 1.1 vol.% for MgZn6.5Ga2 in HT condition [30]). Thus, with an increase in the amount of MgZn and Mg_5_Ga_2_ intermetallic phases, which act as cathodes with respect to α-Mg in the alloy, the cathodic reaction became kinetically easier.

The corrosion potential (*E_corr_*) as a function of extrusion temperature of the alloys is presented in Figure 4a. It was observed that *E_corr_* was a function of Zn content, and for alloys with 2, 4, and 6.5 wt.% Zn, *E_corr_* was approximately −1.55, −1.46, and −1.40 V, respectively. The detailed influence of Zn on *E_corr_* was reported previously for Mg–Zn–Ca alloys [38,39,40]. In addition, the content of Ga and the addition of Y had an insignificant effect on *E_corr_*. Within the uncertainty of ±0.03 V, the extrusion temperature did not affect *E_corr_* of the alloys.

The CR of the Mg–Zn–Ga–(Y) alloys calculated using their corrosion current densities obtained via electrochemical tests are shown in Figure 4b. A good correlation between CR and *E_corr_* was observed, and a higher CR corresponded to a more positive *E_corr_* and vice versa for most of the alloys and extrusion temperatures. The highest CR of ~5 mm/year was observed for the MgZn6.5Ga2 alloy, which was the same value that obtained via immersion corrosion test (Figure 2). The MgZn4Ga4, MgZn4Ga4Y0.5, and MgZn4Ga2 alloys showed a medium CR of 2–4 mm/year because of the lower fraction of the MgZn and Mg_5_Ga_2_ cathodic phases [30]. The minimal CR of 1.1–1.5 mm/year was observed for the MgZn2Ga2 alloy, which has the lowest content of intermetallic phases. No significant relationship was observed between CR and extrusion temperature considering the confidence limit. The CR values obtained via electrochemical tests were several times higher than those obtained from the immersion corrosion tests, which can be attributed to the shielding effect of the corrosion products. Nevertheless, the lowest CR was expected for MgZn2Ga2, which was used for subsequent cytotoxicity tests.

### 3.2. Cytotoxicity

The influence of extraction time on the pH and color of the extract is shown in Figure 5a. To consider the interaction of alloy samples with the microenvironment of human tissues, the samples after the corrosion test in Hanks’ solution were also subjected to cytotoxicity tests. After the first 12 h of extraction, the pH increased from 7.5 to 8.5, and remained constant during further incubation of the samples in the extract. At the same time, the color of the extract changed from red to fuchsia due to the presence of the phenol red indicator in DMEM. Subsequently, the medium became clear, possibly due to the increase in the concentration of metallic ions. The immersion of samples in Hanks’ solution before the extraction process did not affect the pH. The influence of extraction time on the Mg, Zn, and Ga concentrations in the extract is shown in Figure 5b. After the first 72 h, the Mg concentration reached 120 μg/mL and remained constant. The contents of Zn and Ga were lower than that of Mg, and after the first 24 h, a constant concentration of ~3.5 μg/mL was observed for both Zn and Ga. A high content of Ga (>10 μg/mL) was observed after 168 h of extraction, but when samples after immersion in Hanks’ solution were used, the Ga content was lower, perhaps due to the high uncertainty when detecting low concentrations of elements.

The influence of extraction time and extract volume on cell viability after 48 h incubation with MgZn2Ga2 alloy samples extruded at 150 °C is shown in Figure 6a. The 100% extract reduced cell viability by up to 60%. A slight decrease in cell viability was attributed to the high pH of this extract and the high Mg ion concentration [41,42]. After dilution, no less than 85% cell viability was observed for extract fractions of 50%, 25%, and 12.5%. Further, the cell viability increased with increasing extraction time from 12 to 72 h. The reason for this behavior is unknown and requires further investigation. The cell viability after 48 h incubation for MgZn2Ga2 alloy samples extruded at 150 °C and those after immersion in Hanks’ solution prior to extract preparation are shown in Figure 6b. No significant differences in the cell viability were observed for alloy samples immersed in Hanks’ solution compared to the as-prepared ones.

### 3.3. Surface Analysis

Figure 7 shows the microstructures of the corrosion product layers formed on the MgZn6.5Ga2 and MgZn2Ga2 alloys extruded at 150 °C after immersion corrosion tests in Hanks’ solution and incubation in DMEM/F12 cell medium. After the corrosion test, the surface of MgZn6.5Ga2, which showed the highest CR, was covered with deep corrosion cavities up to 340 μm. However, the surface cavities on MgZn2Ga2 (with a low CR) did not exceed 60 μm after the corrosion test. On the MgZn2Ga2 alloy samples after 168 h of incubation in DMEM/F12 extracts, large round corrosion cavities with a moderate depth (~120 μm) were observed. However, when the samples were immersed in Hanks’ solution before extract preparation, cavities with the same depth as those obtained after incubation in DMEM/F12 were observed.

The microstructure and EDS maps of areas with the corrosion product layer formed on MgZn6.5Ga2 and MgZn2Ga2 alloys extruded at 150 °C after immersion tests in Hanks’ solution and incubation in DMEM cell medium are shown in Figure 8. The elemental compositions of the areas designated in Figure 8 as A, B, C, and D are shown in Table 2. After the immersion corrosion tests, the cavities of Mg(OH)_2_, designated as A, and a thin layer of Mg_3_Ca_3_(PO_4_)_4_, designated as B, were visible on the MgZn6.5Ga2 and MgZn2Ga2 alloy surfaces, in accordance with previous works [29,36,43]. When the MgZn2Ga2 alloy sample was subjected to extract preparation in DMEM, the cavity (A) and thin surface layer (B) were also observed, but they were both composed of Mg(OH)_2_. Four distinct areas were observed on the MgZn2Ga2 surface after immersion in Hanks’ solution and further incubation in DMEM/F12. Areas A and B correspond to cavities composed of Mg(OH)_2_, and a slight difference in Ca and P content was observed between them. The surface layer of the corrosion products consisted of a thin Mg_3_Ca_3_(PO_4_)_4_ layer (C) and a thicker Mg(OH)_2_ layer (D). Areas B and C were formed during the corrosion test, and A and D formed during the extract preparation.

## 4. Discussion

Biodegradability is a key requirement for a material that used for biodegradable implants in osteosynthesis. Nevertheless, the excessively high biocorrosion rate, typical for most magnesium alloys, poses a major problem consisting of the high quantity of hydrogen that evolves, owing to a corrosion of Mg in body fluid. This may provide the formation and rise of hydrogen cavities that can hinder bone growth and form a gas gangrene [44,45]. One of the main problems with the use of magnesium implants is they are susceptible to pitting corrosion. It reduces the implant mechanical integrity, and may cause premature failure prior to the accomplishment of the healing process [46]. The biodegradation rate must make sure the biodegradation time is sufficiently long for bone fracture healing. In addition, the healing process must be finished before the mechanical disintegration of the implant. Commonly, the healing process of leg fractures is completed within 3–6 months [47,48,49,50].

The acceptable daily dosage of H_2_ that rat tissues can remove from the body is 0.295 mL/cm^2^ [51]. For most of the investigated alloys, during the first 24 h, the amount of released H_2_ was higher than that value (Figure 1). This could result in possible inflammation of the tissue on the first day after implantation, which is unacceptable in the early postoperative period; therefore, the generated gas should be removed using a subcutaneous needle [46]. Moreover, a protective coating can be applied to minimize the CR and H_2_ release. At the same time, with increasing immersion time and decreasing CR, the H_2_ amount remained within the acceptable range.

The almost neutral pH of the corrosion media for most investigated Mg–Zn–Ga–(Y) alloys in the range of 7.5–8.5 could favor cell viability and proliferation when used as implants [52]. In contrast, the alkaline pH of the corrosion media during the corrosion of MgZn6.5Ga2 alloy (8.9–9.3) could have a negative effect on the healing of bone tissues due to hemolysis [37,53]. It should be noted that in in vivo conditions, the pH value must be different, due to the work of circulatory system in the human body [54].

The CR of the Mg–Zn–Ga–(Y) alloys obtained via electrochemical tests were several times higher than those obtained from the immersion corrosion tests, with the exception of the MgZn6.5Ga2 alloy. The lower CR values obtained via immersion tests can be attributed to the shielding effect of the corrosion products. In accordance with both corrosion test results (electrochemical and immersion corrosion tests), the extrusion temperature had a negligible effect on the CR of the alloys. It is known that CR values obtained from electrochemical tests are affected by the negative difference effect. Therefore, it is better to use immersion corrosion tests to estimate the biodegradation rate of the investigated alloys [32,40,51]. It is known that the implants were subjected to alternating loads, leading to corrosion film spallation and the formation of cracks. In this case the electrochemical corrosion behavior of alloy is also remarkable. Thus, the MgZn2Ga2 alloy has the best corrosion resistance in comparison with other investigated alloys in this work because of the lowest CR obtained in electrochemical corrosion test and one of the lowest CR in accordance with the immersion corrosion test.

Some difficulties in determining the cytotoxicity of biodegradable Mg alloys are observed due to the release of metallic ions, which increase the pH and osmolarity [55]. The pH buffering of extracts by dilution using DMEM can increase the reliability of cytotoxicity results [55]. Wang et al. recommended using 6–10 times extract dilution during the cytotoxicity test to better simulate in vivo environmental conditions [54]. For those dilution levels, no cytotoxicity was observed for the MgZn2Ga2 alloy.

It is known that overdose of Mg is associated with muscular paralysis, hypotension, and respiratory distress, and even leads to cardiac arrest [56]. Zn in high concentrations is neurotoxic and can hinder bone development [46]. Ga overdose can lead to nausea, vomiting, anemia with mild leukopenia, and less often neurologic, pulmonary, and dermatologic effects [57]. Because of that, the release of metallic ions from the implant to tissues must be known and controlled. The normal blood serum level for Mg and Zn are 11.7–25.8 and 0.8–1.1 μg/mL, respectively, which are lower than those obtained in the extracts after recommended 6–10 times dilution [41,55]. Kubásek et al. showed that a Ga concentration of 0.64 μg/mL did not result in cytotoxicity after 48 h of incubation. However, after 5 d of incubation, the cell viability of U-2 osteosarcoma cells was close to that of the phenolic control [25]. The in vivo and extract-preparing conditions are different and because of that the obtained one in the extract concentration of elements must be lower in real in vivo conditions due to the work of the circulatory system in the human body. Overall, the MgZn2Ga2 alloy extruded at 150 °C showed no cytotoxic effect according to ISO 10993-5, because the reduction in cell viability was less than 30%.

Marco et al. compared the corrosion behavior of different Mg alloys in phosphate-buffered saline (PBS) and DMEM, and showed that a clear difference between fast and slow degrading samples is noticeable in PBS. However, in DMEM, no significant difference was observed [58]. In this work it was established that the CR of investigated alloys in DMEM/F12 cell medium was higher than that in Hanks’ solution (Figure 7). The differences in corrosion behavior and composition of the corrosion products obtained for alloys are associated with the different contents of Ca and P in DMEM/F12 and Hanks’ solutions [42]. Presence of Ca and P leads to formation of Mg_3_Ca_3_(PO_4_)_4_ layer which acts as a corrosion barrier. Marco et al. also found that during corrosion in DMEM, a layer of (Mg,Ca)CO_3_ was formed on the surface of the alloy sample. The carbon content is not provided in Table 2 owing to its low repeatability, but for example, in area D in Figure 8d, the carbon content is higher than in other areas, and compared to those of alloys immersed in Hanks’ solution.

It was shown previously that the presence of the corrosion products on an alloy surface affects cell fusion/differentiation and high amount of corrosion products have detrimental effect on the number of osteoclast progenitor cells and the mature osteoclast cell function [59]. The formation of Mg(OH)_2_ and Mg_3_Ca_3_(PO_4_)_4_ corrosion products on the surface of an implant contributes to high osteoblastic activity [50,60,61].

This study was the first attempt to investigate the influence of hot extrusion on the CR and cytotoxicity of newly developed Mg–Zn–Ga–(Y) alloys. The obtained CR for alloys can help to compare the investigated alloy and choose proper alloy composition. However, the implants work in the human body, where both mechanical loading and biocorrosion make an impact on implant lifetime. For a better understanding of the mechanical behavior of the implants, the mechanical properties in the body fluids corrosive environment like corrosion fatigue and stress corrosion cracking it is necessary to known for Mg–Zn–Ga–(Y) alloys. The in vitro biocompatibility of MgZn2Ga2 was approved on the MG63 osteosarcoma cells, but further step is the in-vivo investigation that could allow the application of the developed alloy as biodegradable material.

Further research should focus on stress corrosion cracking and corrosion fatigue of Mg–Zn–Ga–(Y) alloys for the selection of the best composition. In order to decrease the biodegradation rate and gas formation rate the appropriate coating must be developed for increasing of bone integration. Finally, the in vivo test on larger animals to establish the increasing of bone growth rate and improvement of the healing process in comparison with other magnesium alloys must be conducted.

## 5. Conclusions

Five Mg–Zn–Ga–(Y) alloys with different Zn and Ga contents were prepared and subjected to hot extrusion. A low CR in Hanks’ solution was observed for most of the investigated alloys, satisfying the typical requirements for the biodegradation rate of osteosynthesis implants. In accordance with the results of electrochemical and immersion corrosion tests, the CR of the investigated alloys was not dependent on the extrusion temperature. Increasing the Zn content resulted in a higher cathodic current and positive shift of *E_corr_*, responsible for the higher CR. However, increasing the Ga content did not affect the CR.

The high cytotoxicity was obtained only for the extract with no dilution, and the cytotoxicity effect was due to the high pH of the cell medium. Thus, the MgZn2Ga2 alloy showed no cytotoxicity to MG63 cells when the recommended dilutions of extracts are used for cytotoxicity test. Immersion of samples in Hanks’ solution before extract preparation did not affect the cytotoxicity.

The observed differences in corrosion behavior and corrosion-product composition were associated with different Ca and P contents in DMEM/F12 and Hanks’ solutions. We conclude that the extruded MgZn2Ga2 alloys have favorable overall performance for use in bone-fixation implants, considering their CR and biocompatibility.

## Figures and Tables

**Figure 1 jfb-13-00294-f001:**
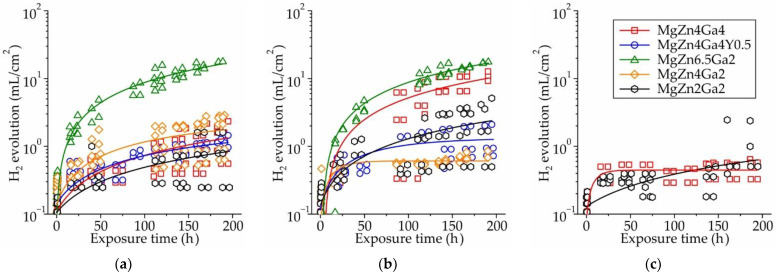
Hydrogen evolution during 192 h of immersion in Hanks’ solution at 37 °C for the Mg–Zn–Ga–(Y) alloys extruded at: (**a**) 150 °C, (**b**) 200 °C, or (**c**) 250 °C.

**Figure 2 jfb-13-00294-f002:**
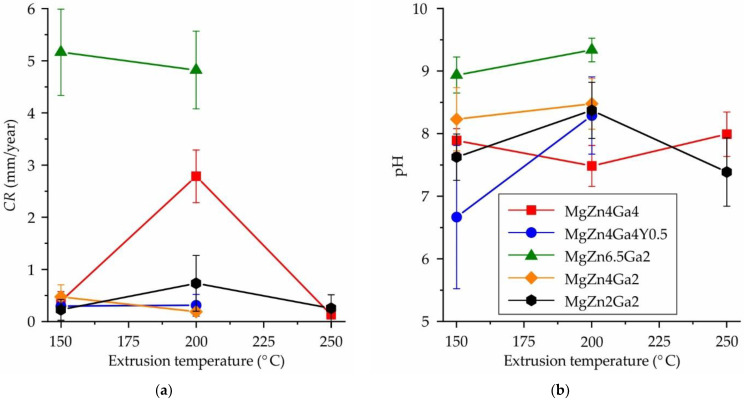
Effect of extrusion temperature and alloy composition on (**a**) corrosion rate (CR) obtained from the immersion tests and (**b**) pH of the corrosion media.

**Figure 3 jfb-13-00294-f003:**
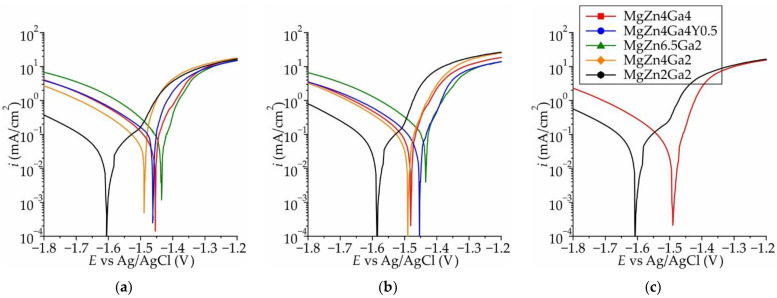
Polarization curves obtained in Hanks’ solution at 37 °C for the Mg–Zn–Ga–(Y) alloys extruded at: (**a**) 150 °C, (**b**) 200 °C, and (**c**) 250 °C.

**Figure 4 jfb-13-00294-f004:**
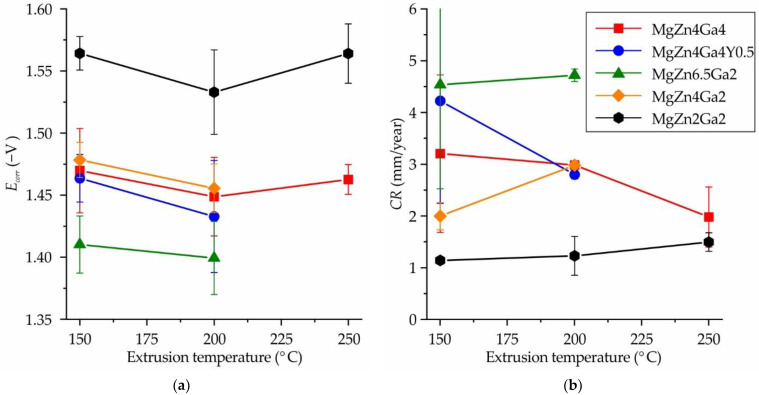
Effects of the extrusion temperature and alloy composition on the (**a**) corrosion potential (*E_corr_*) and (**b**) corrosion rate (CR) obtained via electrochemical corrosion tests.

**Figure 5 jfb-13-00294-f005:**
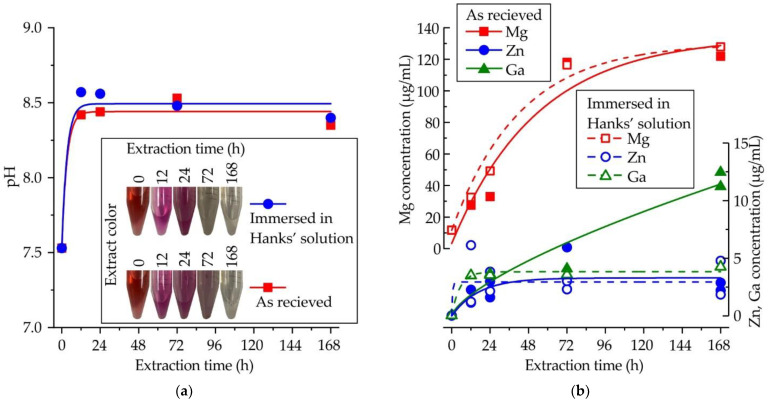
Variation of (**a**) pH and (**b**) contents of Mg, Zn, and Ga in the DMEM/F12 cell medium as a function of extraction time for MgZn2Ga2 alloy extruded at 150 °C.

**Figure 6 jfb-13-00294-f006:**
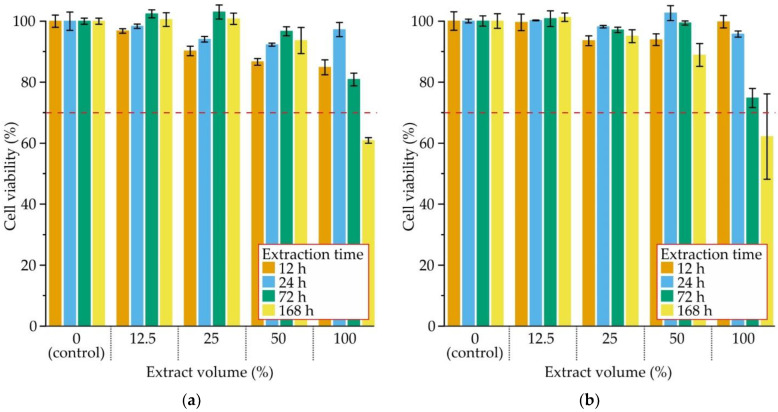
The influence of extract volume and extraction time on cell viability after 48 h incubation for (**a**) as-received samples and (**b**) after 192 h of immersion in Hanks’ solution.

**Figure 7 jfb-13-00294-f007:**
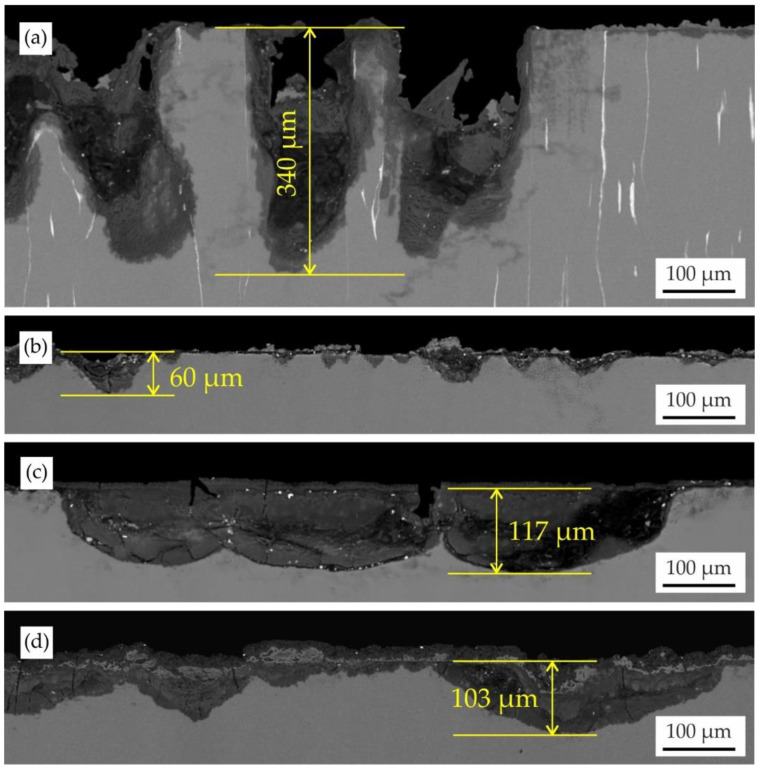
Cross-sectional microstructure of (**a**) MgZn6.5Ga2 and (**b**–**d**) MgZn2Ga2 extruded at 150 °C after: (**a**,**b**) 192 h immersion in Hanks’ solution at 37 °C; (**c**) 168 h incubation in DMEM/F12 cell medium at 37 °C under 5% CO_2_ humidified sterile environment (**d**) after 192 h immersion in Hanks’ solution at 37 °C and 168 h incubation in DMEM/F12 cell medium at 37 °C under 5% CO_2_ humidified sterile environment.

**Figure 8 jfb-13-00294-f008:**
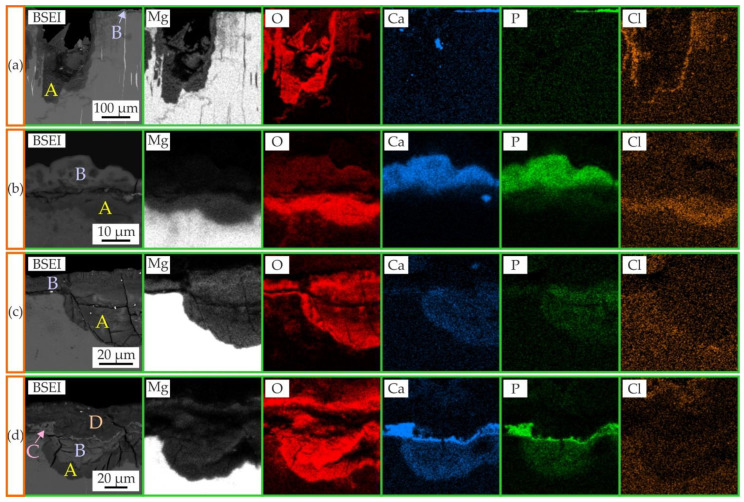
EDS maps showing the distribution of Mg, O, Ca, P, and Cl in (**a**) MgZn6.5Ga2 and (**b**–**d**) MgZn2Ga2 extruded at 150 °C after: (**a**,**b**) 192 h immersion in Hanks’ solution at 37 °C; (**c**) 168 h incubation in DMEM/F12 cell medium at 37 °C and under 5% CO_2_ humidified sterile environment; (**d**) 192 h immersion in Hanks’ solution followed by 168 h incubation in DMEM/F12 cell medium (under the same conditions as the other samples). The composition of areas A, B, C, and D are shown in Table 2.

**Table 1 jfb-13-00294-t001:** Elemental compositions of the prepared alloys.

Alloy	Element Content (wt.%)			
	Mg	Zn	Ga	Y
MgZn4Ga4	Bal.	4.2	4.1	-
MgZn4Ga4Y0.5	Bal.	4.2	4.1	0.4
MgZn6.5Ga2	Bal.	6.5	2.0	-
MgZn4Ga2	Bal.	4.2	2.2	-
MgZn2Ga2	Bal.	2.3	2.3	-

**Table 2 jfb-13-00294-t002:** Surface layer composition (at.%) of MgZn6.5Ga2 and MgZn2Ga2 extruded at 150 °C after immersion corrosion test in Hanks’ solution (37 °C for 192 h) and incubation in DMEM/F12 cell medium (37 °C for 168 h under 5% CO_2_ humidified sterile environment). The areas are defined in Figure 8.

Sample	Area	Element Content (at.%)						
		O	Mg	Zn	Ga	P	Ca	Cl
MgZn6.5Ga2 Hanks’ solution	A	62.5	35.5	0.7	0.2	0	0	0.1
	B	71.4	6.0	0.1	0	10.3	11.3	0
MgZn2Ga2 Hanks’ solution	A	70.6	24.4	0.5	0.5	1.4	0.4	0.9
	B	71.2	4.9	0	0	10.8	12.3	0.1
MgZn2Ga2 DMEM/F12 incubation	A	64.5	28.8	1.0	0.3	1.8	1.9	0.1
	B	66.2	31.8	0.2	0.1	0.1	0.1	0
MgZn2Ga2 Hanks’ solution & DMEM/F12 incubation	A	64.3	24.9	2.4	1.5	1.7	4.5	0
	B	65.3	25.2	1.1	1.0	3.6	2.3	0.2
	C	69.6	12.6	0.1	0.1	9.3	7.6	0
	D	54.3	36.8	0.9	0.7	0.9	1.2	0.2

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
