# Peer review of "Corrosion Behavior and Biocompatibility of Hot-Extruded Mg–Zn–Ga–(Y) Biodegradable Alloys"

_jfb, 2022, doi:10.3390/jfb13040294_

Round 1

Reviewer 1 Report

1. The authors did not explain the novelty and significance of their work in the introduction part. Indeed, the introduction part is not cohesive. Topics change from sentence to sentence. The authors should follow the funnel procedure. The funnel technique for writing the introduction begins with generalities and gradually narrows your focus until you present your thesis.

2. On page 2, line 46, the sentence “In comparison with permanent implants, the biodegradable implants must guarantee the same rates of implant degradation and bone tissue growth for substituting the voids in degraded implant with new bone tissue.” needs the following references: Emerging magnesium-based biomaterials for orthopedic implantation, Emerg. Mater. Res. (2019), pp. 305-319.

3. It is necessary to add in the text information related to the health problems associated with the magnesium, zinc and gallium overdose. It is necessary to present such information since a person receiving an implant of this material may have health problems related to magnesium overdose. The tolerable upper intake level for magnesium at 40 mg per day for adults.

4. Why authors have provided Tafel polarization? It is not discussed in the manuscript.

5. How did the author find Ecorr and Icorr values? For Tafel extrapolation, both anodic and cathodic sides of the curves were considered or only the cathodic side? Authors should provide a brief description for calculating Icorr and Ecorr.

Author Response

  1. The authors did not explain the novelty and significance of their work in the introduction part. Indeed, the introduction part is not cohesive. Topics change from sentence to sentence. The authors should follow the funnel procedure. The funnel technique for writing the introduction begins with generalities and gradually narrows your focus until you present your thesis.

Answer: We try to improve the introduction section of the manuscript.

  1. On page 2, line 46, the sentence “In comparison with permanent implants, the biodegradable implants must guarantee the same rates of implant degradation and bone tissue growth for substituting the voids in degraded implant with new bone tissue.” needs the following references: Emerging magnesium-based biomaterials for orthopedic implantation, Emerg. Mater. Res. (2019), pp. 305-319.

Answer: The reference was added to the manuscript.

  1. It is necessary to add in the text information related to the health problems associated with the magnesium, zinc and gallium overdose. It is necessary to present such information since a person receiving an implant of this material may have health problems related to magnesium overdose. The tolerable upper intake level for magnesium at 40 mg per day for adults.

Answer: This information was added to the article text: “It is known that overdose of Mg is associated with muscular paralysis, hypotension and respiratory distress and even leads to cardiac arrest. Zn in high concentrations is neurotoxic and can hinder bone development. The Ga overdose can lead to nausea, vomiting, anemia with mild leukopenia, and fewer often neurologic, pulmonary, and dermatologic effects.”. It is also reported that some of overdose Mg can be removed by urine.

  1. Why authors have provided Tafel polarization? It is not discussed in the manuscript.

Answer: The discussion of polarization curves is presented: “Typical polarization curves obtained in Hanks' solution at 37 °C for the Mg–Zn–Ga–(Y) alloys extruded at different temperatures are shown in Fig. 3a-c. The polarization curves were similar for all extrusion temperatures. The variations in the anodic Tafel slopes for samples extruded at 200 °C was low and attributed to the different rates of inhibition of the anodic process of electrochemical corrosion, which is associated with the formation of intermetallic compounds [37, 38]. However, the cathodic current differed significantly, and the maximum and minimum cathodic currents were observed for MgZn6.5Ga2 and MgZn2Ga2, respectively. Previously it was shown that with increasing of Zn and Ga content in alloy the summarized fraction of MgZn and Mg5Ga2 phases increased (form 0 vol.% for MgZn2Ga2 to 1.1 for MgZn6.5Ga2 in HT condition [29]). Thus, with an increase in the amount of MgZn and Mg5Ga2 intermetallic phases, which act as cathodes with respect to α-Mg in the alloy, the cathodic reaction became kinetically easier.”

  1. How did the author find Ecorr and Icorr values? For Tafel extrapolation, both anodic and cathodic sides of the curves were considered or only the cathodic side? Authors should provide a brief description for calculating Icorr and Ecorr.

Answer: Electrochemical process parameters, corrosion potential (Ecorr), corrosion current density ( i corr ), anodic and cathodic Tafel slopes, are determined by extrapolation of the Tafel region from polarization curves.

Reviewer 2 Report

This article proposed a systematic study on the corrosion rate and biocompatibility of Mg–Zn–Ga–(Y) alloys prepared by the hot-extrusion. It was verified by various methods that the Mg–2Zn–2Ga alloy has improved bone tissue regeneration ability. It is very interesting and provides a good idea for perspective for applications in osteosynthesis implants.

And there are some questions in the follows.

1.        In the 237th line, “A good correlation between CR and Ecorr was observed, and a higher CR corresponded to a more positive Ecorr and vice versa.” But not all groups are positively correlated. In Figure 4(a) and Figure 4(b), contrasted with 150 °C, MgZn4Ga2 and MgZn2Ga2 in 200°C had higher CR but lower Ecorr.

2.        In the 370h line, “. Thus, the MgZn2Ga2 alloy have the best corrosion resistance in comparison with other investigated alloys in this work because of lowest CR obtained in both electrochemical and immersion corrosion tests.” However, in Figure 2(a), MgZn4Ga4 in 250 °C and MgZn4Ga2 in 200 °C had lower CR.

3.        In the 426h line, “In accordance with the results of electrochemical and immersion corrosion test the CR of investigated alloys was not dependent on the extrusion temperature.” The data of the relation between the CR and the extrusion temperature lacks statistical analysis.

Author Response

  1. In the 237th line, “A good correlation between CR and Ecorrwas observed, and a higher CR corresponded to a more positive Ecorr and vice versa.” But not all groups are positively correlated. In Figure 4(a) and Figure 4(b), contrasted with 150 °C, MgZn4Ga2 and MgZn2Ga2 in 200°C had higher CR but lower Ecorr.

Answer: We agree with the reviewer comment and change this part to: “A good correlation between CR and Ecorr was observed, and a higher CR corresponded to a more positive Ecorr and vice versa for most of alloys and extrusion temperatures.”

  1. In the 370h line, “. Thus, the MgZn2Ga2 alloy have the best corrosion resistance in comparison with other investigated alloys in this work because of lowest CR obtained in both electrochemical and immersion corrosion tests.” However, in Figure 2(a), MgZn4Ga4 in 250 °C and MgZn4Ga2 in 200 °C had lower CR.

Answer: We agree with the reviewer comment. The following changes have been made: “Thus, the MgZn2Ga2 alloy has the best corrosion resistance in comparison with other investigated alloys in this work because of lowest CR obtained in electrochemical corrosion test and one of the lowest CR in accordance with the immersion corrosion test.”

  1. In the 426h line, “In accordance with the results of electrochemical and immersion corrosion test the CR of investigated alloys was not dependent on the extrusion temperature.” The data of the relation between the CR and the extrusion temperature lacks statistical analysis.

Answer: We agree that statistical analysis is helpful in this case, but the confidence limit in most cases is higher than difference between CR for alloys with different grain size. Also, the difference in grain size for alloys is not critical. Maybe CR between the as-cast alloy and extruded alloy can be significant.
